# Unveiling the anatomy of mode-coupling theory

I. Pihlajamaa[1], V. E. Debets[1], C. C. L. Laudicina[1], L. M. C. Janssen[1*]

**1** Soft Matter and Biological Physics, Department of Applied Physics, Eindhoven University of Technology, P.O. Box 513, 5600 MB Eindhoven, Netherlands

*l.m.c.janssen@tue.nl

July 7, 2023

## Abstract

The mode-coupling theory of the glass transition (MCT) has been at the forefront of fundamental glass research for decades, yet the theory's underlying approximations remain obscure. Here we quantify and critically assess the effect of each MCT approximation separately. Using Brownian dynamics simulations, we compute the memory kernel predicted by MCT after each approximation in its derivation, and compare it with the exact one. We find that some often-criticized approximations are in fact very accurate, while the opposite is true for others, providing new guiding cues for further theory development.

# 1 Introduction

Predicting the dynamics of dense and supercooled liquids stands out as one of the largest unsolved problems in classical physics. The most striking feature is that the dynamics exhibit an orders-of-magnitude slowdown as the glass transition is approached, whilst the microstructure remains almost unaltered. Concomitantly, complex two-step relaxation and stretched exponential behavior emerge, structural relaxation becomes increasingly heterogeneous in space and time, and the Stokes-Einstein relation is violated [1, 2]. The mode-coupling theory of the glass transition (MCT) is widely considered to be the only first-principles approach that can describe the dynamics of glass-forming liquids [3–8]. MCT is able to predict the drastic increase of the relaxation time upon supercooling from solely structural information as input, it provides an intuitive mechanism for the slowdown in terms of the cage effect, and it makes precise predictions for the remarkable relation between exponents governing the two-step relaxation process. During the last decades the theory has been successfully applied to a wide range of different systems, e.g. those including confinement [9,10], curved geometries [11], self-propelling particles [12–17], molecular particles [18–22], polymers [23–27], multiple particle species [28–31], external fields [32–34], aging [35, 36], shear [37, 38], high dimensionalities [39–41], and re-entrant phenomena [42, 43].

Even though MCT is by no means an exact theory, it is all-pervasive in theories of the glass transition and arises naturally from many different theoretical approaches and perspectives. For example, it is central to the scenario sketched by random first order transition theory [44], and provides a clear connection between the theories of spin-glasses and structural glasses [45–47]. In particular, schematic MCT is exact for some spin-glasses [45]. In a field-theoretic setting, MCT can be derived as a self-consistent one-loop resummation [46, 48–50], and has been likened to a Landau theory [46, 51] as well as a mean-field theory [40, 44, 52] for the glass transition. Various kinetic-like approaches have also led to the same MCT equations [48, 53].

Despite its successes and ubiquity, MCT is also criticized for failing to capture several key qualitative and quantitative features of glassy dynamics. In particular, it typically overestimates the glassiness of a material to a varying degree which depends on the specific system studied [7,54]. Additionally, the theory does not account for the so-called dynamic crossover [55, 56]. Specifically, MCT predicts that the structural relaxation time scales as a power law with temperature and ultimately diverges at an ideal glass transition. In many simulations and experiments of glass-forming liquids, this power law is indeed also observed, but typically only at mildly supercooled temperatures; instead of diverging, the experimental relaxation time eventually crosses over into an Arrhenius (exponential) scaling [57–60]. The temperature at which this crossover occurs is usually referred to as the mode-coupling temperature $T_{\mathrm{MCT}}$. The inability of MCT to predict the crossover to Arrhenius behavior renders the theory generally only applicable at relatively weak degrees of supercooling [7, 56].

Interestingly, the crossover temperature $T_{\mathrm{MCT}}$, like the theory it is named after, emerges repeatedly as an important and almost universal characteristic temperature for liquid dynamics near the glass transition. In fact, apart from the Kauzmann temperature $T_{\mathrm{K}}$ and the experimental glass transition temperature $T_{\mathrm{g}}$ itself, the mode-coupling temperature is the only one to appear so consistently. Physically, $T_{\mathrm{MCT}}$ is often interpreted as the temperature below which structural reorganizations become dominated by collective 'activated' or 'hopping' events instead of non-cooperative relaxation [1,60–62]. Relatedly, around this crossover temperature, the potential energy landscape manifestly loses all its delocalized unstable modes, suggesting that the crossover is caused by a localization transition [63].

In the random first order transition theory scenario, this is interpreted as a transition to a 'mosaic' of local metastable states [44]. It is thus evident that the breakdown of MCT at $T_{\text{MCT}}$ coincides with a physical change in the behavior of glassy liquids. Consequently, a clear understanding of this breakdown is vital in order to advance towards a more accurate, and ultimately exact theoretical description of the glass transition.

Many attempts have been made to rectify MCT, but these have either been largely fruitless, at least in a qualitative sense, or they have abandoned the first-principles approach in favor of ad hoc corrections to change the predicted scenario. These efforts include (but are not limited to) extended MCT [64,65], generalized MCT [66–69], and its off-diagonal cousin [70,71], and more formal theories [72,73]. There is a large divide between these different approaches, not only in the method by which they attempt to improve upon the theory, but also in the choice of the specific MCT approximations they seek to address. The reason for this disunity of the field is mainly that the various approximations made in the derivation of MCT are notoriously unintuitive, technical, and uncontrolled, rendering it difficult to decide which approximation should be improved in the first place. Given that there is no consensus on which of the MCT approximations should be addressed, it is not surprising that there also exists no agreement on the method by which to do so. Moreover, during the conception of the theory it has been suggested that the MCT approximations should be treated as one entangled set [74]; While this may be understandable from a purely technical point of view, it makes it even more obscure how to move forward.

Here we present a fundamentally new approach to investigate the validity and failures of MCT. Instead of heuristically comparing its predictions with experimental and simulation observations (which has been the approach thus far [29,54,55,75,75–82]) we rigorously disentangle the different MCT approximations made in the derivation of the theory to reveal its inner workings. Arguably this approach has been deemed too challenging in the past due to the technicalities involved, yet here we show that it is now clearly within computational reach. Specifically, we identify and critically assess five different approximations made to the MCT memory kernel: ($i$) neglecting projected dynamics; ($ii$) the projection on density doublets; ($iii$) the diagonalization approximation; ($iv$) the factorization approximation; and ($v$) the convolution approximation. We compute the memory kernel before and after applying each of these approximations directly from simulations of a frequently used model liquid, unambiguously judging their validity. Our approach thus exposes the anatomy of microscopic MCT, allowing us to rule out a complete class of MCT improvements and providing much-needed guidance for the development of a more accurate first-principles theory of the glass transition.

## 2   Exact theory of the dynamics of colloidal liquids

Let us first specify our system of interest. We consider the dynamics of a three-dimensional colloidal fluid of $N$ particles. The position $\mathbf{r}_i$ of particle $i$ evolves according to the overdamped Langevin equation [83]

$$\dot{\mathbf{r}}_i = \zeta^{-1}\mathbf{F}_i + \boldsymbol{\xi}_i(t), \tag{1}$$

in which $\zeta$ is the friction coefficient, $\mathbf{F}_i$ is the potential force acting on particle $i$, and $\boldsymbol{\xi}_i(t)$ is a random force that satisfies $\langle \boldsymbol{\xi}_i(t) \rangle = \mathbf{0}$, and $\langle \boldsymbol{\xi}_i(t) \cdot \boldsymbol{\xi}_j(t') \rangle = 6D_0\delta_{ij}\delta(t-t')$, in which $D_0 = k_B T/\zeta$ is the self-diffusion coefficient. We denote the thermal energy by $k_B T$. For the interaction term we use the repulsive Weeks-Chandler-Andersen potential (see Appendix B.1 for details). In order to assess the MCT approximations in the cleanest

possible test case, we avoid any potentially confounding effects due to polydispersity or non-additivity [75], and hence we focus on a simple monodisperse system in the liquid regime.

The particle trajectories generated by Eq. (1) contain in principle the full dynamics of the system, but one can equivalently consider the joint probability distribution $P(\mathbf{r}_1, \ldots, \mathbf{r}_N, t)$, which specifies the probability density of finding a particle $i$ in a volume $d\mathbf{r}_i$ centered around $\mathbf{r}_i$ at time $t$. In equilibrium, when the probability density is time-independent, $P(\mathbf{r}_1, \ldots, \mathbf{r}_N)$ defines an ensemble average of some observable $A$ as

$$\langle A \rangle = \int d\mathbf{r}_1 \ldots d\mathbf{r}_N A(\mathbf{r}_1, \ldots, \mathbf{r}_N) P(\mathbf{r}_1, \ldots, \mathbf{r}_N). \tag{2}$$

The probability density function formally evolves in time according to the Smoluchowski equation $\dot{P}(\mathbf{r}_1, \ldots, \mathbf{r}_N, t) = \Omega P(\mathbf{r}_1, \ldots, \mathbf{r}_N, t)$, in which $\Omega$ is the Smoluchowski operator

$$\Omega = \sum_{i=1}^{N} \left[ D_0 \nabla_i^2 - \zeta^{-1} \nabla_i \cdot \mathbf{F}_i \right]. \tag{3}$$

This operator will become important when defining the MCT approximations below.

In the context of dense liquids and the glass transition, we are mainly interested in the structural relaxation dynamics of the liquid. A standard probe for such structural relaxation is the intermediate scattering function

$$F(k,t) = \frac{1}{N} \left\langle \rho_{\mathbf{k}}^* e^{\Omega^\dagger t} \rho_{\mathbf{k}} \right\rangle = \frac{1}{N} \left\langle \rho_{\mathbf{k}}^* \rho_{\mathbf{k}}(t) \right\rangle. \tag{4}$$

Here $\rho_{\mathbf{k}} = \sum_j e^{i\mathbf{k} \cdot \mathbf{r}_j}$ is a density mode, i.e. the Fourier transform of the microscopic density at wave vector $\mathbf{k}$ ($k = |\mathbf{k}|$). The operator $\Omega^\dagger$ is the Hermitian conjugate of the Smoluchowski operator, which does not act on the probability distribution $P$ in the definition of the ensemble average. The initial condition of the intermediate scattering function is the static structure factor $S^{(2)}(k) \equiv F(k, t = 0)$, where we have added the superscript (2) to clarify that this is a two-point density correlation function. Note that for isotropic liquids such as the one considered in this work, $F(k,t)$ and $S^{(2)}(k)$ depend only on the magnitude $k$ of the wave vector. Both $F(k,t)$ and $S^{(2)}(k)$ can be readily obtained from scattering experiments or computer simulations and are therefore also widely studied in theories and experiments of dense liquids [84].

In order to obtain an exact equation of motion for the density modes $\rho_{\mathbf{k}}(t)$ and their associated correlation function $F(k,t)$, we use the operator formalism of Mori and Zwanzig [85, 86]. The basic principle is to decompose the space of dynamical variables into a resolved subspace, which is spanned by the density modes, and an unresolved subspace containing all other dynamical variables. Briefly, we perform this decomposition by introducing a projector $\mathcal{P} = \rho_{\mathbf{k}} \langle \rho_{\mathbf{k}}^* \rho_{\mathbf{k}} \rangle^{-1} \langle \rho_{\mathbf{k}}^*$ that projects onto the space spanned by the density modes, and the associated orthogonal projector $\mathcal{Q} = 1 - \mathcal{P}$. For technical reasons unique to Brownian systems (elaborated in Appendix A), we also need a second exact projection step with the projectors $\mathcal{P}' = \rho_{\mathbf{k}} \langle \rho_{\mathbf{k}}^* \Omega^\dagger \rho_{\mathbf{k}} \rangle^{-1} \langle \rho_{\mathbf{k}}^* \Omega^\dagger$ and $\mathcal{Q}' = 1 - \mathcal{P}'$. The framework enables us to write a generalized Langevin equation for the density modes such that the only coupling between the resolved and the unresolved space is contained in the so-called fluctuating force $R_{\mathbf{k}}(t)$ and a memory kernel $K(t)$ that describes the time-autocorrelation function of the fluctuating force. By multiplying the resulting equation with $\rho_{\mathbf{k}}^*(t)$ and taking an ensemble average, we find the following equation of motion for the intermediate scattering function [8, 87, 88]:

$$\frac{\partial F(k,t)}{\partial t} + \frac{D_0 k^2}{S(k)} F(k,t) + \int_0^t \mathrm{d}\tau K(k, t-\tau) \frac{\partial F(k,\tau)}{\partial \tau} = 0. \tag{5}$$

Here, the memory kernel is defined as

$$K(k,t) = \frac{1}{N k^2 D_0} \left\langle R_{\mathbf{k}}^* e^{\Omega^\dagger \mathcal{Q}'' t} R_{\mathbf{k}} \right\rangle. \tag{6}$$

In this representation, $\mathcal{Q}'' = \mathcal{Q}\mathcal{Q}'$, and

$$R_{\mathbf{k}} = \mathcal{Q}\Omega^\dagger \rho_{\mathbf{k}} = -D_0 k^2 \rho c^{(2)}(k)\rho_{\mathbf{k}} + \frac{i}{\zeta} \sum_j (\mathbf{k} \cdot \mathbf{F}_j) e^{i\mathbf{k}\cdot\mathbf{r}_j} \tag{7}$$

is the fluctuating force, in which $c^{(2)}(k)$ is the direct correlation function [84].

Equation (5) provides an exact description of the dynamics of a dense liquid. Explicitly, if we would be able to compute the exact memory kernel $K(k,t)$, the exact density correlation function $F(k,t)$ can be obtained. However, the kernel poses a major theoretical bottleneck, since there exists no general theory that allows for an exact prediction of $K(k,t)$. It is the aim of mode-coupling theory to approximate this memory kernel such that it can be evaluated in a self-consistent manner. The first difficulty in treating $K(k,t)$ lies in the fact that the exact kernel evolves according to a different differential equation than standard observables, that is, it evolves with $e^{\Omega^\dagger \mathcal{Q}'' t}$ instead of the standard $e^{\Omega^\dagger t}$. This means that it is non-trivial to compute it either from theory or from standard particle-based simulations [89, 90].

Nonetheless, it is possible to write an exact integral equation for the memory kernel by using the Dyson operator identity [91]. The result is

$$K(k,t) = K_{\Omega^\dagger}(k,t) + \int_0^t d\tau K(k, t-\tau) W(k,\tau) \tag{8}$$

in which both $K_{\Omega^\dagger}(k,t)$ and $W(k,\tau)$ are functions that evolve with standard Brownian dynamics and can thus be measured directly from simulations. Their precise definitions are given in Appendix A. Once these functions are measured, we can numerically solve the integral equation of Eq. (8) to find the exact memory kernel $K(k,t)$ governing the full dynamics. This exact kernel will serve as our benchmark in order to assess the quality of the various MCT approximations.

As a consistency check, we have verified our procedure for obtaining the exact memory kernel by inserting the calculated $K(k,t)$ from Eq. (8) back into Eq. (5). The resulting $F(k,t)$ can then be compared to $F(k,t)$ measured directly from the same simulations. This comparison is made in Fig. 1, where the full lines are the direct measurement and the dashed lines are the solutions of Eqs. (8) and (5) at the location of the main peak of the structure factor. These results show that the intermediate scattering function is indeed very faithfully recovered by our procedure, confirming that the obtained $K(k,t)$ is a very accurate reconstruction of the exact memory kernel and thus an accurate benchmark for MCT. Numerical details of this procedure are presented in Appendix B.2.

## 3    Approximations of the memory kernel

Having established the exact equation of motion and the exact memory kernel, we now proceed to assess the validity of various approximations made to the memory kernel within the framework of MCT. In order to do so, we follow the standard MCT derivation [3, 8]

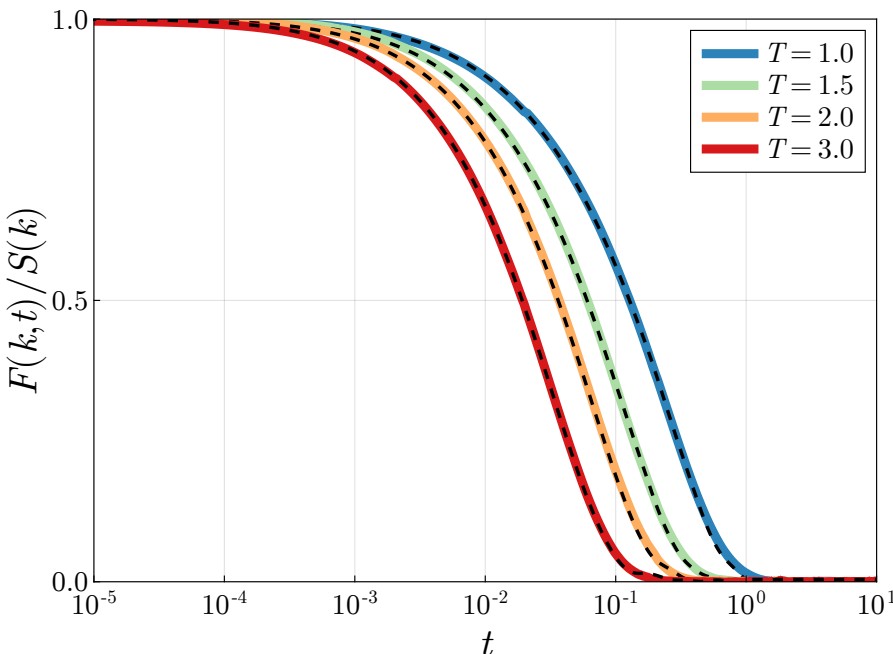

Figure 1: Intermediate scattering functions $F(k,t)$ for our colloidal liquid as a function of time $t$ for different temperatures, evaluated at the location of the main peak of the static structure factor ($k = 7.0$). The solid lines correspond to $F(k,t)$ obtained from direct simulation measurements, while the black dashed lines correspond to the numerical solutions of the exact equations (5) and (8). This comparison serves to validate our numerical procedure to extract the exact memory kernel via Eq. (8).

and evaluate the memory kernel after each main step in the derivation. For completeness we also present the full derivation of MCT in Appendix A. Our key result is presented in Fig. 2, which shows the obtained approximate memory kernels, as well as the corresponding intermediate scattering functions, for the highest and lowest temperatures considered in this work. All comparisons are made at the wave number corresponding to the main peak of the static structure factor, i.e. $k = 7.0$. This wave number is chosen as it corresponds to the typical distance between nearest neighbors, and hence to the typical cage size; Within MCT, this length scale governs the cage effect and is deemed the most important for structural relaxation [92]. In the next subsections, we discuss each of the MCT approximations in order of appearance in the derivation.

## 3.1 Neglecting projected dynamics

The exact memory kernel $K(k,t) = \frac{1}{Nk^2 D_0} \left\langle R_{\mathbf{k}}^* e^{\Omega^\dagger \mathcal{Q}'' t} R_{\mathbf{k}} \right\rangle$ is propagated in time using the operator $e^{\Omega^\dagger \mathcal{Q}'' t}$. Unfortunately, the presence of the orthogonal projector $\mathcal{Q}''$ renders the time evolution of the memory kernel physically non-intuitive and mathematically intractable, since it does not behave in accordance to the same physical laws that underlie the normal Brownian dynamics of microscopic observables (which evolve with $e^{\Omega^\dagger t}$). There exists some analytical work for simple systems expanding $e^{\Omega^\dagger \mathcal{Q}'' t}$ in polynomials of $\mathcal{Q}''$, which thus can be applied to provide increasingly accurate expressions for the memory kernel [93, 94]. However, within mode-coupling theory, the approximation $e^{\Omega^\dagger \mathcal{Q}'' t} = e^{\Omega^\dagger (1 - \mathcal{P}'') t} \approx e^{\Omega^\dagger t}$ is employed to keep the theory tractable. We refer to

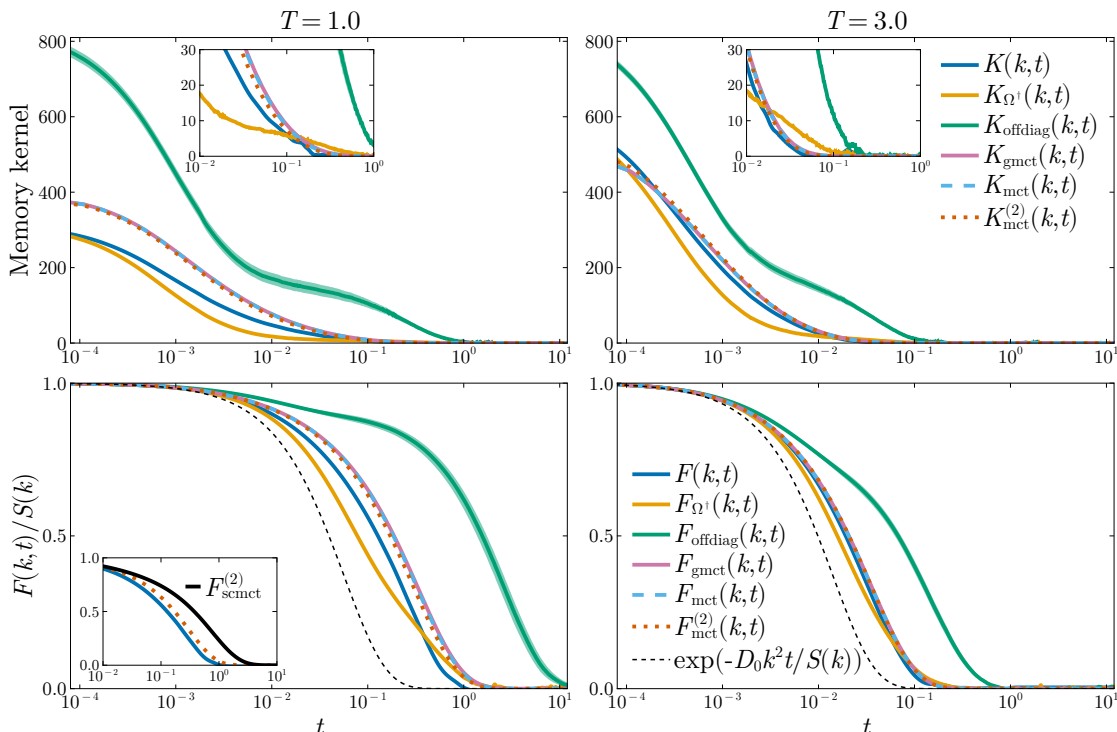

Figure 2: The memory kernel and associated intermediate scattering functions for our colloidal liquid at several steps in the derivation of mode-coupling theory. The top panels show the memory kernels at $T = 1.0$ (left) and $T = 3.0$ (right) as a function of time. Here, $K$ is the exact memory kernel, $K_{\Omega^\dagger}$ is the kernel with projected dynamics, $K_{\text{offdiag}}$, $K_{\text{gmct}}$, $K_{\text{mct}}$, and $K^{(2)}_{\text{mct}}$ are the memory kernels after the doublet projection, diagonalization, factorization, and convolution approximations, respectively. For the meaning of the error bar in the off-diagonal kernel, we refer to the Appendix. The inset is a zoom-in of the final relaxation behavior. The bottom panels show the intermediate scattering functions at the same temperatures as a function of time according to each of the different memory kernels, obtained by solving the corresponding generalized Langevin equation. The black dashed line indicates the exponential relaxation that corresponds to a liquid with no memory. In the bottom left figure, we show in the inset the intermediate scattering functions obtained from $K(t)$, $K^{(2)}_{\text{mct}}$, and from a full self-consistent solution, denoted as $F^{(2)}_{\text{scmct}}$, of the mode-coupling equations using only structure as input.

this approximation as neglecting the projected dynamics. Note that the neglect of $\mathcal{P}''$ in the propagator is also trivially required after the final MCT approximation is made (see Sec. 3.4) [3]. In the present work, however, we treat the neglect of $\mathcal{P}''$ as the first explicit MCT approximation, as it can be imposed separately from later MCT approximations. This first step implies that $K(k,t) \approx K_{\Omega^\dagger}(k,t)$ where $K_{\Omega^\dagger}(k,t)$ is the same function that appears in the integral equation of Eq. (8) for the exact memory kernel.

Figure 2 shows both the exact memory kernel $K(k,t)$ and the approximate kernel without projected dynamics, $K_{\Omega^\dagger}(k,t)$, at the wave number corresponding to the main peak of the static structure factor. It is clear that when $t \to 0$, the two kernels become equal. While this is mathematically trivial, it can also be physically understood by the realization that the fluctuating force $R_\mathbf{k}$ resides in the subspace orthogonal to the density

modes, implying that projections onto the orthogonal subspace have no effect when applied directly. However, as $t$ increases, the influence of the part of the fluctuating force that evolves into the space spanned by density modes grows, resulting in a slower initial decay of the true memory kernel compared to the one with standard Smoluchowski dynamics. The results is that $K_{\Omega^\dagger}(k, t)$ corresponds to more liquefied short-time dynamics than $K(k, t)$.

Interestingly, the relaxation of the projected kernel $K_{\Omega^\dagger}$ at long times is in fact slower than that of the exact memory kernel, developing a very low shoulder that delays the final relaxation process. This can be seen most clearly in the insets of Fig. 2. The corresponding intermediate scattering functions $F_{\Omega^\dagger}(k, t)$ reveal the same pattern, i.e. after an initially faster decay, they ultimately relax over a longer time scale than the exact $F(k, t)$ at both temperatures. This clearly implies that the orthogonal dynamics are not simply "rescaled" regular dynamics, and that the neglect of $\mathcal{P}''$ thus introduces a non-trivial modification of the full time-dependent dynamics.

## 3.2   Projection on density doublets

The next step in the derivation of mode-coupling theory is to project the memory kernel on the space spanned by doublets of density modes. The motivation for this projection is that, next to the singlet density modes $\rho_{\mathbf{k}}$ which we have explicitly included as our resolved variables in the Mori-Zwanzig formalism, the most important dynamic observables for structural relaxation are products of two density modes $\rho_{\mathbf{k}}\rho_{\mathbf{k}'}$ [95]. As we have not included them directly in the theory, their effects must still be contained within the memory kernel, and can thus be extracted by projecting it on the space of all density doublets $\rho_{\mathbf{q}}\rho_{\mathbf{q}'}$. After carrying out this projection, we obtain what we call the off-diagonal memory kernel, which contains only contributions originating from the space of density doublets,

$$K_{\text{offdiag}}(k, t) = \frac{\rho^2 D_0}{4N^3} \sum_{\mathbf{q}'\mathbf{q}} V_{\mathbf{k},\mathbf{q}'}^* \left\langle \rho_{\mathbf{q}'}^* \rho_{\mathbf{k}-\mathbf{q}'}^* \rho_{\mathbf{q}}(t)\rho_{\mathbf{k}-\mathbf{q}}(t) \right\rangle V_{\mathbf{k},\mathbf{q}}. \tag{9}$$

Note that here the density modes evolve in time with normal dynamics $e^{\Omega^\dagger t}$. In the above, we have introduced the vertex as

$$V_{\mathbf{k},\mathbf{q}} = \frac{N}{2ik\rho D_0} \sum_{\mathbf{q}''} \left\langle \rho_{\mathbf{q}}^* \rho_{\mathbf{k}-\mathbf{q}}^* \rho_{\mathbf{q}''} \rho_{\mathbf{k}-\mathbf{q}''} \right\rangle^{-1} \left\langle \rho_{\mathbf{q}''}^* \rho_{\mathbf{k}-\mathbf{q}''}^* R_{\mathbf{k}} \right\rangle, \tag{10}$$

which can be interpreted as a static coupling constant for wave vectors $\mathbf{k}$ and $\mathbf{q}$. The inverse four-point structure factor

$$\left(S^{(4)}\right)^{-1} (\mathbf{q}, \mathbf{k} - \mathbf{q}, \mathbf{q}'', \mathbf{k} - \mathbf{q}'')/N^3 \equiv \left\langle \rho_{\mathbf{q}}^* \rho_{\mathbf{k}-\mathbf{q}}^* \rho_{\mathbf{q}''} \rho_{\mathbf{k}-\mathbf{q}''} \right\rangle^{-1} \tag{11}$$

appears here as the normalization of the density-doublet projector such that it is idempotent.

While the time-dependent off-diagonal four-point density correlation function in Eq. (9) can be evaluated numerically, its static inverse is unfortunately more problematic. The reason is that it is defined by the relation

$$2 \sum_{\mathbf{q}_1 \mathbf{q}_2} \left\langle \rho_{\mathbf{k}_1}^* \rho_{\mathbf{k}_2}^* \rho_{\mathbf{q}_1} \rho_{\mathbf{q}_2} \right\rangle^{-1} \left\langle \rho_{\mathbf{q}_1}^* \rho_{\mathbf{q}_2}^* \rho_{\mathbf{k}_3} \rho_{\mathbf{k}_4} \right\rangle = \delta_{\mathbf{k}_1 \mathbf{k}_3} \delta_{\mathbf{k}_2 \mathbf{k}_4} + \delta_{\mathbf{k}_1 \mathbf{k}_4} \delta_{\mathbf{k}_2 \mathbf{k}_3}, \tag{12}$$

rendering the problem of finding it an intractably large linear algebra problem. To proceed, we therefore simplify the vertices by neglecting the off-diagonal terms, retaining only the

terms in the sum for which $\mathbf{q}'' = \mathbf{q}$ and $\mathbf{q}'' = \mathbf{k} - \mathbf{q}$, i.e.,

$$V_{\mathbf{k},\mathbf{q}} \approx \frac{N}{ik\rho D_0} \left\langle \rho_{\mathbf{q}}^* \rho_{\mathbf{k}-\mathbf{q}}^* \rho_{\mathbf{q}} \rho_{\mathbf{k}-\mathbf{q}} \right\rangle^{-1} \left\langle \rho_{\mathbf{q}}^* \rho_{\mathbf{k}-\mathbf{q}}^* R_{\mathbf{k}} \right\rangle \tag{13}$$

$$= (\hat{\mathbf{k}} \cdot \mathbf{q}) c^{(2)}(\mathbf{q}) + \hat{\mathbf{k}} \cdot (\mathbf{k} - \mathbf{q}) c^{(2)}(\mathbf{k} - \mathbf{q}) - k\rho c^{(3)}(-\mathbf{k}, \mathbf{k} - \mathbf{q}, \mathbf{q}), \tag{14}$$

where we have introduced the direct correlation functions $c^{(2)}$ and $c^{(3)}$. These can be related to the corresponding structure factors as $1/S^{(2)}(k) = 1 - \rho c^{(2)}(k)$, and

$$\frac{S^{(3)}(\mathbf{k}_1, \mathbf{k}_2, \mathbf{k}_3)}{S^{(2)}(\mathbf{k}_1) S^{(2)}(\mathbf{k}_2) S^{(2)}(\mathbf{k}_3)} = 1 + \rho^2 c^{(3)}(\mathbf{k}_1, \mathbf{k}_2, \mathbf{k}_3). \tag{15}$$

Here, the three-point structure factor is defined as $S^{(3)}(\mathbf{k}_1, \mathbf{k}_2, \mathbf{k}_3) = \langle \rho_{\mathbf{k}_1} \rho_{\mathbf{k}_2} \rho_{\mathbf{k}_3} \rangle / N$.

It is important to note that we have now made two independent approximations in this step: the projection on density doublets and the diagonalization of the inverse four-point structure factor in the vertex. We currently have no direct means to separate the effects of these two approximations. Fortunately, however, there is a way in which we can indirectly estimate the validity of the static diagonalization approximation in isolation, namely by considering the $t = 0$ limit of the dynamic four-point correlations. We shall revisit this point in Sec. 4.

To assess the overall quality of this step, we measure the off-diagonal kernel of Eq. (9) with the approximated vertices of Eq. (14) from our simulation data. The results in Fig. 2 clearly show that this step causes a significant overestimation of the memory, resulting in an error in the relaxation time of an order of magnitude. The size of this error seems to increase as the temperature is lowered, suggesting that the discrepancy might become more severe as the glass transition is approached. Moreover, the shoulder that is already visible in the memory kernel $K_{\Omega^\dagger}$ is much more pronounced in the off-diagonal kernel. The presence of this shoulder suggests that the relaxation of the off-diagonal memory kernel consists of a slow and a fast relaxation process, where the slow process is spuriously causing an overestimation of the structural relaxation time.

## 3.3   Diagonalization

Up to this point, we have expressed the approximate memory kernel in terms of static system properties $\rho$, $D_0$, $S^{(2)}(k)$, and $S^{(3)}(\mathbf{k}_1, \mathbf{k}_2, \mathbf{k}_3)$, and a time-dependent part given by the off-diagonal four-point correlation function

$$F^{(4)}(\mathbf{k}_1, \mathbf{k}_2, \mathbf{k}_3, \mathbf{k}_4, t) = \left\langle \rho_{\mathbf{k}_1}^* \rho_{\mathbf{k}_2}^* \rho_{\mathbf{k}_3}(t) \rho_{\mathbf{k}_4}(t) \right\rangle / N. \tag{16}$$

To proceed, there are two possible approaches. Firstly, one may construct a separate equation of motion for $F^{(4)}$, which can be solved self-consistently with Eq. (5). This approach is called off-diagonal generalized mode-coupling theory [70,71]. The main drawback of this idea is that the integrals involved are difficult to evaluate numerically within reasonable computational time due to the large combinatorial space of wave vector arguments. The second approach is much more common [96] and is also used in classical mode-coupling theory: from the four-point function $F^{(4)}$, all off-diagonal terms are neglected. Specifically, we keep only two diagonal terms in the sum of Eq. (9), that is, the terms where $\mathbf{q}' = \mathbf{q}$ and $\mathbf{q}' = \mathbf{k} - \mathbf{q}$. Thus, upon diagonalization only those remain and all other terms vanish:

$$K_{\text{gmct}}(k, t) = \frac{\rho^2 D_0}{2N^3} \sum_{\mathbf{q}} |V_{\mathbf{k},\mathbf{q}}|^2 \left\langle \rho_{\mathbf{q}}^* \rho_{\mathbf{k}-\mathbf{q}}^* \rho_{\mathbf{q}}(t) \rho_{\mathbf{k}-\mathbf{q}}(t) \right\rangle. \tag{17}$$

We stress that, similar to the diagonalization of the inverse four-point structure factor $\left(S^{(4)}\right)^{-1}$ in the vertices, this technical approximation is uncontrolled.

The reason we choose to denote this memory kernel with the subscript 'gmct' is that this is the same kernel that appears in the first equation of the Generalized Mode-Coupling Theory (GMCT) hierarchy [67–69]. GMCT attempts to improve on MCT by retaining the diagonal four-point function explicitly and constructing a new equation of motion for it (i.e. avoiding factorization, which shall be treated in the next step, Section 3.4). Briefly, GMCT proceeds by re-applying the Mori-Zwanzig formalism using density doublets as the resolved variables and projecting the new fluctuating force on density triplets, yielding a six-point density correlation function in the new memory kernel. In principle this scheme can be continued for arbitrarily many density modes, creating an infinite hierarchy which can be truncated or solved self-consistently at arbitrary finite order. In this way, GMCT seeks to delay the factorization approximation of MCT. We note that in some works, the dynamic diagonalization approximation and factorization are collectively referred to as "the factorization approximation", because the factorization of a four-point function implies its diagonalization. For clarity we keep them separate here.

Figure 2 shows that the diagonalization approximation of the dynamic four-point density correlation has a major effect on the predicted memory kernel: compared to the off-diagonal kernel of the previous step, the kernel is reduced by approximately a factor of two, and the time scale of relaxation is about an order of magnitude faster. This large effect of the diagonalization approximation can be seen as a confirmation that the approximation is inherently uncontrolled, but at the same time it also partially corrects for the significant overestimation error introduced in the previous step. Note also that the clear shoulder present in the off-diagonal kernel has disappeared, suggesting that the two relaxation processes seen in the decay of the off-diagonal memory can in fact be identified as a fast process characterized by diagonal density decorrelations and a slow off-diagonal contribution.

## 3.4  Factorization

The next and sometimes last step in the derivation of classical mode-coupling theory is to factorize the diagonal four-point function in terms of a product of two-point functions,

$$K_{\text{mct}}(k,t) = \frac{\rho^2 D_0}{2N^3} \sum_{\mathbf{q}} |V_{\mathbf{k},\mathbf{q}}|^2 \left\langle \rho_{\mathbf{q}}^* \rho_{\mathbf{q}}(t) \right\rangle \left\langle \rho_{\mathbf{k}-\mathbf{q}}^* \rho_{\mathbf{k}-\mathbf{q}}(t) \right\rangle \tag{18}$$

$$= \frac{\rho^2 D_0}{2N} \sum_{\mathbf{q}} |V_{\mathbf{k},\mathbf{q}}|^2 F(q,t) F(|\mathbf{k}-\mathbf{q}|,t). \tag{19}$$

We denote this memory kernel with the subscript 'mct' since it is the standard kernel that is widely used in microscopic mode-coupling theory [5]. Notably, this kernel is expressed in terms of the intermediate scattering functions $F$, and hence Eq. (5) now becomes a self-consistent equation. In order to test the factorization approximation, however, we do not solve Eqs. (5) and (18) self-consistently, but rather we evaluate the kernel of Eq. (18) directly from simulation data, similar to how the previous memory kernels were computed.

From Fig. 2 it can be seen that the data of $K_{\text{gmct}}$ are identical to those of $K_{\text{mct}}$ within our error margins. This forces us to conclude that, at least in the liquid regime, the factorization approximation of diagonal density correlations is very accurate and can be employed without caution. We have confirmed that $\left\langle \rho_{\mathbf{k}_1}^* \rho_{\mathbf{k}_2}^* \rho_{\mathbf{k}_1}(t) \rho_{\mathbf{k}_2}(t) \right\rangle$ and $\left\langle \rho_{\mathbf{k}_1}^* \rho_{\mathbf{k}_1}(t) \right\rangle \left\langle \rho_{\mathbf{k}_2}^* \rho_{\mathbf{k}_2}(t) \right\rangle$ show similar agreement in our simulations.

The validity of the factorization approximation is, in fact, not very surprising, since there exists a host of literature (e.g. [97]) showing that the four-point dynamic suscep-

tibility $\chi^{(4)}(\mathbf{k}_1, \mathbf{k}_2, t) = \langle \rho^*_{\mathbf{k}_1} \rho^*_{\mathbf{k}_2} \rho_{\mathbf{k}_1}(t) \rho_{\mathbf{k}_2}(t) \rangle - \langle \rho^*_{\mathbf{k}_1} \rho_{\mathbf{k}_1}(t) \rangle \langle \rho^*_{\mathbf{k}_2} \rho_{\mathbf{k}_2}(t) \rangle$ scales as $\mathcal{O}(N)$ [1] whereas the two terms on the right hand side scale with $\mathcal{O}(N^2)$. The notion that relative fluctuations are vanishingly small in the thermodynamic limit is typical in statistical physics. Since the fluctuations captured in $\chi^{(4)}$ are a direct measure for the error of the factorization approximation, one can readily infer that in the thermodynamic limit, the factorization approximation becomes *exact*. This statement holds throughout the supercooled phase as long as $\chi^{(4)}$ remains finite, which simulations indicate it does [97–99] (mode-coupling theory predicts it to diverge only at the ideal glass transition, but to remain finite everywhere else [32]). In this light it is hard to justify attempts to avoid or delay the factorization of four-point density correlations in cases where one is willing to diagonalize them.

### 3.5 Convolution approximation

The last approximation usually employed in the derivation of MCT is the convolution approximation for the vertices [100], which simplifies the required static input for the theory. Although there are analytical results for the three-point direct correlation function $c^{(3)}$ of hard particles [101], the theory becomes more tractable if it only requires two-point functions as input. To this end, the convolution-approximation is often employed, setting $c^{(3)} = 0$ [102] (see [31, 103, 104] for notable exceptions). We add a superscript (2) to the mode-coupling theory memory kernel $K_{\mathrm{mct}}(k, t)$ to indicate that structural triplet correlations are neglected. Note that we could also have made this approximation at any earlier point in the theory, but we conjecture that the effect of it is insensitive to when it is actually employed.

We show in Fig. 2 that the neglect of triplet correlations in the vertices only has a small quantitative effect on the dynamics, very weakly increasing or decreasing the predicted memory kernel and intermediate scattering functions depending on the temperature. Note that the lines for $K_{\mathrm{gmct}}(k, t)$, $K_{\mathrm{mct}}(k, t)$, and $K^{(2)}_{\mathrm{mct}}(k, t)$ lie very close together and are therefore hard to distinguish. This is clear evidence that the inclusion of triplet correlations is unnecessary for our colloidal liquid at the density and temperatures considered in this work.

The fact that the triplet correlations have no significant influence on the predicted dynamics indicates that, at least in the liquid regime studied in this work, the microscopic structure is still well described by two-point correlation functions only. In general, however, the validity of the convolution approximation depends highly on the material and state point studied. For example, the incorporation of triplet correlations is known to be more important for strong network-forming systems than for fragile models such as the one studied here [104], and supercooling to lower temperatures may also give rise to nontrivial higher-order structural features [105–110].

## 4 Discussion

In this work we have explicitly resolved each of the main approximations comprising the standard mode-coupling theory of the glass transition, allowing us to unveil the effect of each consecutive approximation on the predicted memory kernel for a model colloidal liquid. In all cases, the approximate kernel could be benchmarked against the exact result, providing an unambiguous test for the theory's validity. Let us now discuss the

---

[1] In fact, $\chi_4$ can be obtained by taking two functional derivatives of the free energy with respect to intensive fields. This means that $\chi_4$ must extensive.

relative importance of each MCT approximation step. Our main results, summarized in Fig. 2, clearly show that, apart from the factorization and convolution approximations, all approximations have a significant and non-trivial effect on the memory kernel. Specifically, the first three approximations affect both the absolute magnitude of the kernel and the time scales of decay, biasing the predictions either towards more liquid-like or more glassy dynamics and imposing qualitatively different decay patterns. Curiously, after the final MCT approximation is made, the predicted intermediate scattering function is closest to the exact dynamics, at least in the regime of dense (yet not supercooled) liquids studied in this work. We also point out that the MCT approximations which are virtually exact, i.e. factorization and convolution, are ironically the ones for which the most attempts have been made in the past to circumvent them.

Our work identifies two key approximations that manifestly impact the memory kernel the most, both of which involve neglecting off-diagonal four-point density correlations. Explicitly, when going from $K_{\Omega^\dagger}$ to $K_{\text{offdiag}}$ we neglect the *static* off-diagonal terms in the vertices, and when going from $K_{\text{offdiag}}$ to $K_{\text{gmct}}$ we neglect the *dynamic* off-diagonal terms. These two steps coincide with a significant increase and decrease of the approximate memory kernel, respectively. However, recall that the diagonalization of the static four-point function only applies to its *inverse* $\left(S^{(4)}\right)^{-1}$ [see Eq. (14)], whereas for the dynamic case the diagonalization approximation is applied to the standard correlation function. We surmise that this causes the two diagonalization approximations to have, in effect, opposite signs. The combined result of both approximations is thus a fortuitous cancellation of errors, which we believe also underlies, at least in part, the success of standard MCT.

As already mentioned in the introduction, it is well known that standard MCT has the general tendency to overestimate the glassiness of a system. Our results of Fig. 2 now allow us to expose precisely which step in the MCT derivation is responsible for this overestimation, namely $K_{\text{offdiag}}$. Recall that at this step the projection onto density doublets is introduced, combined with our diagonalization of $\left(S^{(4)}\right)^{-1}$. Unfortunately, we are currently unable to directly separate the effects of these two approximations due to the great computational difficulties associated with evaluating the off-diagonal version of $\left(S^{(4)}\right)^{-1}$. However, we can provide indirect evidence that the static diagonalization, rather than the projection on doublets itself, is the more likely cause of the overestimation error of $K_{\text{offdiag}}$. Briefly, we find that the diagonalization of the *dynamic* four-point function introduces a change in the predicted memory kernel of around 50% at $t \to 0$ (comparing $K_{\text{offdiag}}$ with $K_{\text{gmct}}$ at $T = 1.0$). We expect that employing this same approximation to the inverse four-point correlation function in each vertex should introduce at least a similar error in the opposite direction (going from $K_{\Omega^\dagger}$ to $K_{\text{offdiag}}$). This is also consistent with what we observe in Fig. 2. Strengthened by the fact that the projection on density doublets itself is sometimes claimed to be exact [68][2], we conjecture that the main source of error in MCT ultimately stems from the neglect of off-diagonal density correlations.

In contrast to the method by which we have evaluated the MCT memory kernel, which is to use the intermediate scattering function $F$ obtained from simulations, the usual way to solve MCT is to do so self-consistently. That is, the two-point correlation function $F$ that appears in $K_{\text{mct}}$ is chosen such that it satisfies equation (5) with $K_{\text{mct}}$ as memory kernel. This self-consistency effectively magnifies the error made by MCT, since any small error propagates iteratively through both the kernel and $F$ itself. From the main results in Fig. 2 we can already infer that self-consistent MCT should yield an overestimation of the intermediate scattering function as compared to our directly calculated $F_{\text{mct}}$. To see this, we write $K_{\text{mct}} \equiv K_{\text{mct}}[F]$ and note from our results that $F_{\text{mct}} > F$ for all times, at least

---

[2]Even if the projection on doublets were not exact, it should be very accurate in systems with sufficiently high temperatures such that structural two-body correlations dominate higher order ones.

at low temperatures. It follows that $K_{\text{mct}}[F_{\text{mct}}] > K_{\text{mct}}[F]$, and hence the overestimation error will be further increased in subsequent self-consistent iterations until convergence is reached. To numerically confirm that this is indeed the case, we show the self-consistent MCT solution $F^{(2)}_{\text{scmct}}$ in the inset of the bottom left panel of Fig. 2. Explicitly, for our system at $T = 1$, self-consistent MCT predicts a relaxation time of $\tau_\alpha = 0.6$, whereas our measurements give $\tau_\alpha = 0.3$ for MCT, and the true relaxation time is only $\tau_\alpha = 0.2$. In addition to this overestimation, the self-consistency property also gives rise to the prediction of a spurious divergence of the relaxation time. Overall our main results thus understate the severity of the errors made by self-consistent MCT.

Using our results, we have argued that there is little reason to attempt to improve MCT by delaying or avoiding the factorization approximation specifically. Nevertheless, such attempts have had significant success in recent years in the form of (diagonal) generalized mode-coupling theory, as it typically improves upon the quantitative predictions of MCT [67–69, 111, 112]. In hindsight, we believe that these improvements are again fortuitous consequences of another cancellation-of-errors effect. In order to solve the equation of motion for the diagonalized four-body correlator $\langle \rho_{\mathbf{k}}^* \rho_{\mathbf{q}}^* \rho_{\mathbf{k}}(t) \rho_{\mathbf{q}}(t) \rangle$, several additional approximations are made within GMCT, whose effects seem to partially cancel the errors made by the standard MCT, yielding quantitatively improved results. If, within GMCT, an exact equation of motion for diagonal four-body correlators in terms of the intermediate scattering function was employed, the theory would reproduce the factorization approximation in the thermodynamic limit and therefore the results of GMCT would be equivalent to that of MCT.

# 5    Conclusion and Outlook

In conclusion, we have unveiled the effect of each of the approximations that enter the mode-coupling theory derivation. Our results explicitly show that the success of standard MCT is rooted in a remarkable cancellation of errors, as conjectured earlier from a different perspective [113]. We have found that the diagonalization approximation in the statics and dynamics has the most significant impact on the predicted dynamics. It is clear from our results that any attempt to improve this approximation by including off-diagonal density correlations should treat both the statics and dynamics on an equal footing lest the predictions of the theory may be worsened.

In future research we aim to apply our methods to a more supercooled system in order to evaluate whether our conclusions hold when the glass transition is approached. Preliminary results suggest that they do. Similarly, it is still an open question to what degree our findings depend on the type of dynamics studied and on the fragility of the material in question. We believe that more work in this direction should inform a more systematic approach to improving one of the most promising theories of the glass transition.

One can envision several routes toward a more quantitative dynamical theory of the glass transition. Firstly, the effects of the projected dynamics can be dealt with rigorously in a self-consistent manner; Indeed, it is not hard to express $W(k, t)$ directly in terms of $F(k, t)$ and its derivatives, which can then be used in conjunction with Eq. (8) to include the effects of projected dynamics. Secondly, in order to include off-diagonal four-point correlations into the theory, novel numerical schemes may be employed, especially efficient integration and inversion routines, to evaluate the off-diagonal memory kernel. To alleviate the computational costs, one may also seek to restrict the full wave vector space to only the most important off-diagonal correction terms. In this regard, including e.g. only the wave vectors corresponding to the main peaks of $S^{(4)}$ [110] might already provide a reasonable

improvement. Finally, formal perturbative diagrammatic corrections to MCT have been derived [48, 72]. Numerical integration or analytic analysis of the diagrams involved may provide invaluable new insights in the microscopic dynamics of the glass transition.

## Acknowledgements

It is a pleasure to thank Andrés Montoya Castillo for stimulating discussions that have inspired parts of this work and Thomas Voigtmann for valuable discussions on the results.

### Author contributions

I. Pihlajamaa: Conceptualization, Methodology, Software, Data Curation, Investigation, Writing: Original Draft & Review
V.E. Debets: Conceptualization, Writing: Review
C.C.L. Laudicina: Conceptualization, Writing: Review
L.M.C. Janssen: Conceptualization, Writing: Review, Supervision, Funding Acquisition

### Funding information

The Dutch Research Council (NWO) is acknowledged for financial support through a Vidi grant (IP, CCLL, and LMCJ) and START-UP grant (VED and LMCJ).

## A    The memory kernel equation

In order to write an equation of motion for the intermediate scattering function $F(k, t)$, we separate the evolution of it into a part that propagates in the space spanned by the density modes, and a space orthogonal to it. This we do by multiplying by $1 = \mathcal{P} + \mathcal{Q}$, in which $\mathcal{P} = \rho_{\mathbf{k}} \rangle \langle \rho_{\mathbf{k}}^* \rho_{\mathbf{k}} \rangle^{-1} \langle \rho_{\mathbf{k}}^*$ is the projector onto the space spanned by the density modes, and $\mathcal{Q} = 1 - \mathcal{P}$ its orthogonal complement. This yields

$$\frac{\partial F(k, t)}{\partial t} = \frac{1}{N} \left\langle \rho_{\mathbf{k}}^* \Omega^\dagger (\mathcal{P} + \mathcal{Q}) e^{\Omega^\dagger t} \rho_{\mathbf{k}} \right\rangle \tag{20}$$
$$= -\frac{k^2 D_0}{S(k)} \left( F(k, t) + \int_0^t d\tau K_{\mathrm{red}}(k, t - \tau) F(k, \tau) \right)$$

where we have used the Dyson decomposition identity,

$$e^{\Omega^\dagger t} = e^{\Omega^\dagger \mathcal{Q} t} + \int_0^t d\tau e^{\Omega^\dagger \mathcal{Q}(t-\tau)} \Omega^\dagger \mathcal{P} e^{\Omega^\dagger \tau} \tag{21}$$

and introduced the reducible memory kernel $K_{\mathrm{red}}(k, t) = \frac{1}{k^2 N D_0} \left\langle R_{\mathbf{k}}^* e^{\mathcal{Q} \Omega^\dagger \mathcal{Q} t} R_{\mathbf{k}} \right\rangle$. Here $R_{\mathbf{k}} = \mathcal{Q} \Omega^\dagger \rho_{\mathbf{k}}$ is the fluctuating force.

Because Eq. (20) is difficult to work with numerically, it is customary to perform a second projection with projector operator $\mathcal{P}' = \rho_{\mathbf{k}} \rangle \langle \rho_{\mathbf{k}}^* \Omega^\dagger \rho_{\mathbf{k}} \rangle^{-1} \langle \rho_{\mathbf{k}}^* \Omega^\dagger$ in order to change the form of the time integral into one that is more stable [87]. After invoking the Dyson decomposition identity again in the form

$$e^{\mathcal{Q} \Omega^\dagger \mathcal{Q} t} = e^{\mathcal{Q} \Omega^\dagger \mathcal{Q}' \mathcal{Q} t} + \int_0^t d\tau e^{\mathcal{Q} \Omega^\dagger \mathcal{Q}(t-\tau)} \mathcal{Q} \Omega^\dagger \mathcal{P}' \mathcal{Q} e^{\mathcal{Q} \Omega^\dagger \mathcal{Q}' \mathcal{Q} \tau}, \tag{22}$$

we find

$$K_{\text{red}}(k,t) = K(k,t) - \int_0^t \mathrm{d}\tau K_{\text{red}}(k, t-\tau)K(k,\tau), \tag{23}$$

in which $K(k,t) = \frac{1}{Nk^2 D_0} \left\langle R_\mathbf{k}^* e^{\Omega^\dagger \mathcal{Q}' \mathcal{Q} t} R_\mathbf{k} \right\rangle$ is the irreducible memory kernel. In Laplace space, Eqs. (20) and (23) can be straightforwardly combined to give in real space

$$\frac{\partial F(k,t)}{\partial t} + \frac{D_0 k^2}{S(k)} F(k,t) + \int_0^t \mathrm{d}\tau K(k,t-\tau)\frac{\partial F(k,\tau)}{\partial \tau} = 0. \tag{24}$$

In the main text, we have introduced $\mathcal{Q}'' \equiv \mathcal{Q}'\mathcal{Q}$ to simplify notation.

To confirm that we have not yet made any approximations, we can use the Dyson identity one last time,

$$e^{\Omega^\dagger \mathcal{Q}' \mathcal{Q} t} = e^{\Omega^\dagger t} - \int_0^t \mathrm{d}\tau e^{\Omega^\dagger \mathcal{Q}' \mathcal{Q}(t-\tau)} \Omega^\dagger (1 - \mathcal{Q}'\mathcal{Q}) e^{\Omega^\dagger t}, \tag{25}$$

to find an integral equation for the irreducible memory kernel, yielding

$$K(k,t) = K_{\Omega^\dagger}(k,t) + \int_0^t d\tau K(k,t-\tau)W(k,\tau) \tag{26}$$

in which we introduce the fluctuating force auto-correlation function

$$K_{\Omega^\dagger}(k,t) = (Nk^2 D_0)^{-1} \left\langle R_\mathbf{k}^* e^{\Omega^\dagger t} R_\mathbf{k} \right\rangle \tag{27}$$

and the correlation function between the fluctuating force and $\Omega^\dagger \rho_\mathbf{k}$,

$$W(k,t) = (Nk^2 D_0)^{-1} \left\langle \rho_\mathbf{k}^* \Omega^\dagger e^{\Omega^\dagger t} R_\mathbf{k} \right\rangle. \tag{28}$$

Since the latter two quantities evolve in accordance with the standard evolution operator $e^{\Omega^\dagger t}$, we can measure them directly from particle resolved Brownian dynamics simulations. The results are shown in Fig. 3. The first of the two can be rewritten as

$$K_{\Omega^\dagger}(k,t) = \frac{1}{N} \left( \frac{\beta}{\zeta k^2} \left\langle h_\mathbf{k}^* e^{\Omega^\dagger t} h_\mathbf{k} \right\rangle - 2\frac{\rho c(k)}{\zeta} \left\langle \rho_\mathbf{k}^* e^{\Omega^\dagger t} h_\mathbf{k} \right\rangle + D_0 k^2 \left( \rho c(k) \right)^2 \left\langle \rho_\mathbf{k}^* e^{\Omega^\dagger t} \rho_\mathbf{k} \right\rangle \right), \tag{29}$$

in which we have introduced $h_\mathbf{k} = i\sum_j (\mathbf{k}\cdot\mathbf{F}_j)e^{i\mathbf{k}\cdot\mathbf{r}_j}$. We denote this quantity as $K_{\Omega^\dagger}$ seeing that it is equivalent to the irreducible memory kernel $K$ evolving with the Smoluchowski operator $\Omega^\dagger$ instead of with $\Omega^\dagger \mathcal{Q}'\mathcal{Q}$. Similarly, for $W(k,t)$ we can write

$$W(k,t) = \frac{1}{N} \left( \frac{\beta}{\zeta k^2} \left\langle h_\mathbf{k}^* e^{\Omega^\dagger t} h_\mathbf{k} \right\rangle - \frac{1 + \rho c(k)}{\zeta} \left\langle \rho_\mathbf{k}^* e^{\Omega^\dagger t} h_\mathbf{k} \right\rangle + D_0 k^2 \rho c(k) \left\langle \rho_\mathbf{k}^* e^{\Omega^\dagger t} \rho_\mathbf{k} \right\rangle \right). \tag{30}$$

The latter two equations show that these functions have many terms in common (in fact, their definitions differ only by a $\mathcal{Q}$), which is reflected in their remarkably similar decay as presented in Fig. 3(c,d).

Now that we have measured the functions $W(k,t)$ and $K_{\Omega^\dagger}(k,t)$, we can solve the integral equation Eq. (8) numerically, and insert the result into Eq. (5). The intermediate scattering function found by this method can be compared to one directly measured from the same simulations. This comparison is made in Fig. 1.

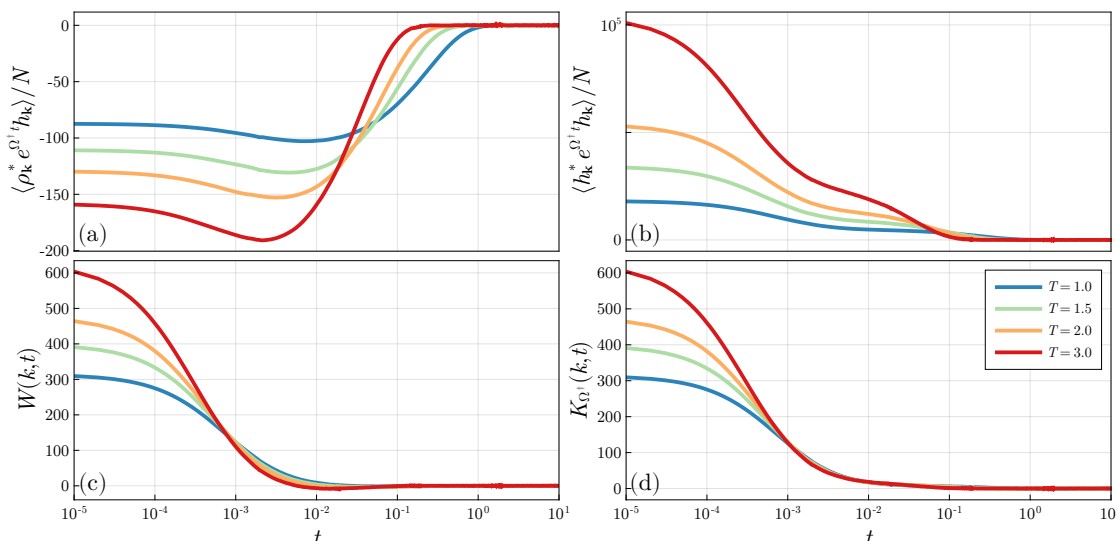

Figure 3: Time-correlation functions as a function of time $t$ at the peak of the static structure factor $k\sigma = 7.0$ obtained by direct simulation measurement. Panel (a) shows the correlation between the density modes $\rho_{\mathbf{k}}$ and the momentum averaged longitudinal stress $h_{\mathbf{k}}$, and (b) displays the auto-correlation function of that stress. In (c) and (d) we show $W(k,t)$ and $K_{\Omega^\dagger}(k,t)$ which together determine the irreducible memory kernel as expressed by Eq. (8) of the main text. The data are extracted from Brownian dynamics simulations of a Weeks-Chandler-Andersen system at number density $\rho = 0.95$ for four different temperatures above the crystallization transition.

In order to derive the mode-coupling equation, we follow the main text and start with the irreducible memory kernel $K(k,t) = \frac{1}{Nk^2 D_0} \left\langle R_{\mathbf{k}}^* e^{\Omega^\dagger \mathcal{Q}' \mathcal{Q} t} R_{\mathbf{k}} \right\rangle$. As a first step, we replace the orthogonal dynamics with standard dynamics, yielding

$$K_{\Omega^\dagger}(k,t) = \frac{1}{Nk^2 D_0} \left\langle R_{\mathbf{k}}^* e^{\Omega^\dagger t} R_{\mathbf{k}} \right\rangle. \tag{31}$$

The second step is to project on density doublets,

$$K_{\text{offdiag}}(k,t) = \frac{1}{Nk^2 D_0} \left\langle R_{\mathbf{k}}^* \mathcal{P}_2 e^{\Omega^\dagger t} \mathcal{P}_2 R_{\mathbf{k}} \right\rangle, \tag{32}$$

in which the projection operator $\mathcal{P}_2$ projects on the space of all density doublets, i.e.

$$\mathcal{P}_2 = \frac{1}{4} \sum_{\mathbf{k}_1 \mathbf{k}_2 \mathbf{k}_3 \mathbf{k}_4} \rho_{\mathbf{k}_1} \rho_{\mathbf{k}_2} \rangle \left\langle \rho_{\mathbf{k}_1}^* \rho_{\mathbf{k}_2}^* \rho_{\mathbf{k}_3} \rho_{\mathbf{k}_4} \right\rangle^{-1} \left\langle \rho_{\mathbf{k}_3}^* \rho_{\mathbf{k}_4}^* \right. . \tag{33}$$

The prefactor is included to prevent over-counting. Substituting (33) into (32), we find

$$K(k,t) \approx \frac{1}{16Nk^2 D_0} \sum_{\mathbf{k}_i} \sum_{\mathbf{k}_i'} \langle R_{\mathbf{k}}^* \rho_{\mathbf{k}_1} \rho_{\mathbf{k}_2} \rangle \left\langle \rho_{\mathbf{k}_1}^* \rho_{\mathbf{k}_2}^* \rho_{\mathbf{k}_3} \rho_{\mathbf{k}_4} \right\rangle^{-1} \left\langle \rho_{\mathbf{k}_3}^* \rho_{\mathbf{k}_4}^* e^{\Omega^\dagger t} \rho_{\mathbf{k}_1'} \rho_{\mathbf{k}_2'} \right\rangle$$
$$\times \left\langle \rho_{\mathbf{k}_1'}^* \rho_{\mathbf{k}_2'}^* \rho_{\mathbf{k}_3'} \rho_{\mathbf{k}_4'} \right\rangle^{-1} \left\langle \rho_{\mathbf{k}_3'}^* \rho_{\mathbf{k}_4'}^* R_{\mathbf{k}} \right\rangle. \tag{34}$$

Inserting the definition of the Smoluchowski operator and integrating by parts, it is

not hard to show that

$$\left\langle \rho_{\mathbf{k}_1}^* \rho_{\mathbf{k}_2}^* R_{\mathbf{k}} \right\rangle = -D_0 N \delta_{\mathbf{k},\mathbf{k}_1+\mathbf{k}_2} \tag{35}$$
$$\times \left[ (\mathbf{k}_1 \cdot \mathbf{k}) S^{(2)}(k_2) + (\mathbf{k}_2 \cdot \mathbf{k}) S^{(2)}(k_1) - \frac{k^2 S^{(3)}(-\mathbf{k}_1, -\mathbf{k}_2, \mathbf{k}_1+\mathbf{k}_2)}{S^{(2)}(k)} \right].$$

This leads us to define the vertex

$$V_{\mathbf{k},\mathbf{q}} = \frac{N^2}{2\rho} \sum_{\mathbf{p}} \left\langle \rho_{\mathbf{q}}^* \rho_{\mathbf{k}-\mathbf{q}}^* \rho_{\mathbf{p}} \rho_{\mathbf{k}-\mathbf{p}} \right\rangle^{-1} \tag{36}$$
$$\times \left[ (\hat{\mathbf{k}} \cdot \mathbf{p}) S^{(2)}(|\mathbf{k}-\mathbf{p}|) + (\hat{\mathbf{k}} \cdot (\mathbf{k}-\mathbf{p})) S^{(2)}(p) - k \frac{S^{(3)}(-\mathbf{p}, \mathbf{p}-\mathbf{k}, \mathbf{k})}{S^{(2)}(k)} \right],$$

so that the memory kernel reduces to

$$K_{\text{offdiag}}(k,t) = \frac{1}{4} \frac{D_0 \rho^2}{N^3} \sum_{\mathbf{q}\mathbf{q}'} V_{\mathbf{k},\mathbf{q}'}^* \left\langle \rho_{\mathbf{q}'}^* \rho_{\mathbf{k}-\mathbf{q}'}^* e^{\Omega^\dagger t} \rho_{\mathbf{q}} \rho_{\mathbf{k}-\mathbf{q}} \right\rangle V_{\mathbf{k},\mathbf{q}}. \tag{37}$$

Next, we neglect all the off-diagonal elements of the inverse four-point structure factor in the vertex, keeping only $\mathbf{p} = \mathbf{q}$ and $\mathbf{p} = \mathbf{k} - \mathbf{q}$:

$$V_{\mathbf{k},\mathbf{q}} \approx \frac{N^2}{\rho} \left\langle \rho_{\mathbf{q}}^* \rho_{\mathbf{k}-\mathbf{q}}^* \rho_{\mathbf{q}} \rho_{\mathbf{k}-\mathbf{q}} \right\rangle^{-1} \tag{38}$$
$$\times \left[ (\hat{\mathbf{k}} \cdot \mathbf{q}) S^{(2)}(|\mathbf{k}-\mathbf{q}|) + (\hat{\mathbf{k}} \cdot (\mathbf{k}-\mathbf{q})) S^{(2)}(q) - k \frac{S^{(3)}(-\mathbf{q}, \mathbf{q}-\mathbf{k}, \mathbf{k})}{S^{(2)}(k)} \right].$$

This approximation is fully uncontrolled. We then factorize the diagonal inverse structure factor into the product of two two-point functions, yielding

$$V_{\mathbf{k},\mathbf{q}} \approx \frac{1}{\rho} S^{(2)}(\mathbf{q})^{-1} S^{(2)}(|\mathbf{k}-\mathbf{q}|)^{-1} \tag{39}$$
$$\times \left[ (\hat{\mathbf{k}} \cdot \mathbf{q}) S^{(2)}(|\mathbf{k}-\mathbf{q}|) + (\hat{\mathbf{k}} \cdot (\mathbf{k}-\mathbf{q})) S^{(2)}(q) - k \frac{S^{(3)}(-\mathbf{q}, \mathbf{q}-\mathbf{k}, \mathbf{k})}{S^{(2)}(k)} \right]$$
$$= (\hat{\mathbf{k}} \cdot \mathbf{q}) c^{(2)}(q) + \hat{\mathbf{k}} \cdot (\mathbf{k}-\mathbf{q}) c^{(2)}(|\mathbf{k}-\mathbf{q}|) - k\rho c^{(3)}(-\mathbf{q}, \mathbf{q}-\mathbf{k}, \mathbf{k}). \tag{40}$$

As we discuss in the main text, this step is exact in the thermodynamic limit. Lastly, we do a convolution approximation of $S^{(3)}$, neglecting $c^{(3)}$, and find

$$V_{\mathbf{k},\mathbf{q}} \approx (\hat{\mathbf{k}} \cdot \mathbf{q}) c^{(2)}(q) + \hat{\mathbf{k}} \cdot (\mathbf{k}-\mathbf{q}) c^{(2)}(|\mathbf{k}-\mathbf{q}|), \tag{41}$$

where we have used the definition of the direct correlation function $1/S^{(2)}(k) = 1 - \rho c^{(2)}(k)$. To arrive at the final mode-coupling theory equation, the dynamical off-diagonal memory kernel can be diagonalized and factorized as indicated in the main text.

# B    Numerical Details

## B.1    Brownian Dynamics Simulations

The results from this work are obtained from trajectories of Brownian dynamics simulations performed with the LAMMPS software package [114]. We use a purely repulsive

single-component system of Weeks-Chandler-Andersen type, characterized by the pair interaction potential

$$U(r) = 4\epsilon \left[ \left(\frac{\sigma}{r}\right)^{12} - \left(\frac{\sigma}{r}\right)^{6} \right] + \epsilon \tag{42}$$

for $r/\sigma < 2^{1/6}$ and $U(r) = 0$ for all other $r$. Here, $r$ is the inter-particle distance, $\sigma = 1$ describes the particle size and $\epsilon = 1$ is the interaction strength. All results are presented in terms of these units. The simulations contain $N = 2000$ particles confined within a cubic box with number density $\rho = 0.95$ and periodic boundary conditions. At the lowest temperature studied, $k_B T = 1$, this system has a liquid-solid coexistence region for $\rho \in (0.96, 1.03)$ [115], which we just stay below.

We integrate the Brownian equations of motion, Eq. (1), using a time step of $\Delta t = 10^{-5}$ and friction coefficient $\zeta = 1$. First, we equilibrate the system for $10^7$ time steps and we subsequently run an equal number of steps for production. During the production run, we save the particle positions on a quasi-logarithmic grid in order to compute the time-dependent quantities. For each of the 4 temperatures studied, we run 50 independent simulations, allowing us to take proper ensemble averages.

## B.2 The exact memory kernel

In order to solve the integral equation (8) to find the exact memory kernel, we first obtain $K_{\Omega^\dagger}(k, t)$ and $W(k, t)$ at $k = 7.0$ from the simulation trajectories. To do so, we straightforwardly evaluate their definitions, Eqs. (29) and (30), whereby we average over all 50 independent simulation trajectories, over all allowed wave vectors in the range $k \in (7.0 \pm 0.1)$, and over a small number of time origins. Because the evaluation of Eq. (8) is highly sensitive to noise, we additionally apply a locally estimated scatterplot smoothing (LOESS) filter with polynomial degree 2 and a smoothing parameter of 0.1 [116]. The resulting smoothed functions are inserted in a discretized version of Eq. (8), for which we have used a non-equidistant Simpson's rule on a logarithmic grid [117]. The memory kernel is subsequently found by solving the resulting system of equations.

To validate the obtained memory kernel, we insert it into Eq. (5), which is solved by the method presented by Fuchs and coworkers [118]. The resulting intermediate scattering function is compared with the measured one in Fig. 1.

## B.3 Off-diagonal memory kernel

To simplify the computation of the off-diagonal memory kernel, Eq. (9), we write it as

$$K_{\text{offdiag}}(k, t) = \frac{\rho^2 D_0}{4N^3} \left\langle B^*(\mathbf{k}, 0) B(\mathbf{k}, t) \right\rangle, \tag{43}$$

in which $B(\mathbf{k}, t) = \sum_{\mathbf{q}} \rho_{\mathbf{q}}(t) \rho_{\mathbf{k}-\mathbf{q}}(t) V_{\mathbf{k},\mathbf{q}}$. We compute this function $B(\mathbf{k}, t)$ at the peak of the static structure factor for all $t$ by explicitly performing the sum over all allowed wave vectors $\mathbf{q}$ up to some cutoff $k_c$. The auto-correlation function of the result yields the off-diagonal memory kernel. We have found that the cutoff required for convergence of this memory kernel is much larger than that needed for diagonal memory kernels. In particular, the cutoff chosen for this memory kernel is temperature dependent and given by $k_c = 83.0, 84.7, 86.2, 88.3$ for $T = 1.0, 1.5, 2.0, 3.0$, respectively. These values are obtained by computing the off-diagonal memory kernel as a function of this cutoff from one set of simulation trajectories and performing a sinusoidal fit to the data. This procedure is illustrated in Fig. 4. The one-sided error bars in Fig. 2 are set equal to the amplitude of the fitted sine wave, providing an overestimation of the convergence error in the off-diagonal memory kernel.

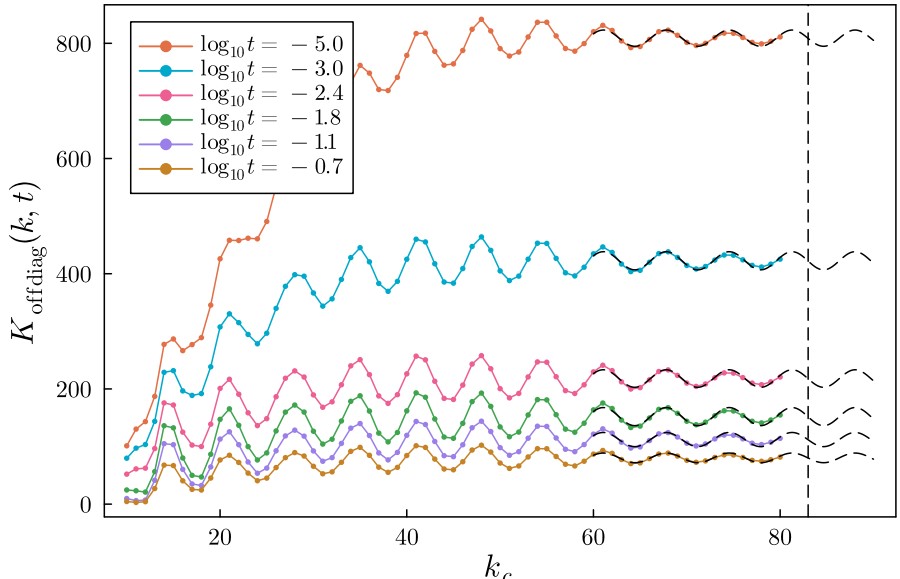

Figure 4: The off-diagonal memory kernel as a function of the cutoff value in the sum from the definition of $B(\mathbf{k}, t)$ for different values of $t$ at $T = 1.0$. The high $k_c$ data is sinusoidally fitted to estimate the maximal convergence error. The vertical dashed line indicates the cutoff used in this work.

The triplet correlation function $c^{(3)}(\mathbf{k}, \mathbf{q})$ appearing in the vertices contributes significantly only for small values $k$ and $q$ [102]. Therefore we set it equal to zero for values of $q > 10.0$, saving computation time and decreasing the amount of noise. For simulations closer to the glass transition, where triplet correlations may play a more dominant role, it might be necessary to increase this cutoff, or forgo it completely. We use the same cutoff for the triplet correlations in the diagonalized kernels.

### B.4    GMCT and MCT memory kernels

The diagonal memory kernels $K_{\mathrm{gmct}}$, $K_{\mathrm{mct}}$, and $K_{\mathrm{mct}}^{(2)}$ converge faster than the off-diagonal one. This allows us to decrease the cutoff wave number of the sums in their definitions to $k_c = 40.0$, equal to that used in many numerical implementations of the standard MCT equations. The convergence error decreases as $t$ increases, because the intermediate scattering function, and thereby the integrand, decays as $F(k, t) \sim e^{-D_0 k^2 t}$ for small $t$ and large $k$. We estimate that at $t = 10^{-3}$ and $t = 10^{-2}$, the relative convergence errors are at most 5% and 0.2%, respectively. For smaller $t$, the error is larger, but the influence of the memory kernel on the dynamics at such short time scales is negligible.

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
