# Peer review of "Unveiling the anatomy of mode-coupling theory"

_SciPost Physics_

## Round 1 · Referee Report · Saroj Nandi (Referee 1) · 2023-10-2

Strengths

  1. This is an important problem for the field of glassy dynamics.

  2. All the details are given such that a broad audience can follow.

  3. The motivation of the problem is clearly written.

Weaknesses

  1. Numerical results are explored in the high-temperature regime.

  2. Citation is not balanced, specific references are missing (see report).

  3. Conclusions are speculative at times.

Report

I have gone through the manuscript titled "Unveiling the anatomy of mode-coupling theory" by Pihlajamaa et al. In this work, the authors explore the effects of various approximations involved in one of the particular derivations of the mode-coupling theory (MCT) that is via the projection operator formalism. MCT plays an important role in the glassy dynamics of equilibrium systems. It has also been extended for nonequilibrium and biological systems. However, the theory enjoys a mixed fate, it predicts a spurious non-ergodicity transition and the theory breaks down beyond this transition. Understanding the reasons behind this breakdown of the theory is an important problem. This manuscript seeks to address this specific problem and therefore, should be generally interesting to a broad audience interested in glassy dynamics. I think this is a competent work and the manuscript is mostly well-written. However, I have several questions and comments that I would like the authors to address before I can recommend it for publication.

(1) I think the authors should rewrite the first sentence.

(2) The introductory paragraph provides a list of problems where MCT has been successfully applied. This shows that the theory has some of the right elements to describe the glassy dynamics and it is important to understand the reasons behind the failure of the theory. Yet, some of the recent important applications of MCT are missing from this list: For example, for self-propelled particles: Nandi and Gov, Soft Matter (2017), Paul et al (PNAS 2023) have applied the active MCT for length scales in these systems. MCT for the two and four-point susceptibilities has been applied to an aging system in Nandi and Ramaswamy, Phys Rev Lett (2012). Also, I think it is important to mention the confluent systems as a very recent example of a near-ideal system for MCT [https://arxiv.org/abs/2306.07250] and some of the author's own works.

(3) I am confused by this statement: "It is thus evident that the breakdown of MCT at TMCT coincides with a physical change ... ". Note that all the examples are indirect. Presenting these as physical examples can be misleading. Although I do agree with the authors that "a clear understanding of this breakdown is vital".

(4) Last paragraph, Sec 1: The authors should mention that there are many different ways to derive the MCT. The projection operator formalism is one of them, although it might be one of the most popular ones. This work focuses on the approximations within this derivation though some of the steps are related with other derivations too. I think focussing on a particular derivation is fine, but it should be mentioned.

(5) Last paragraph, Sec 2: The results presented in Fig. 1 and discussed in this paragraph can be confusing. It gives the impression that the reason MCT fails is due to the set of approximations made later in the theory and Eq. (5) and (8) agree exactly with the simulation data. This is a strong claim but Fig. 1 itself cannot support this. The simulations are not yet in the supercooled regime and deviation already started being significant for T=1.0 (the lowest temperature reported in this work). I think the deviation will grow wider at even lower temperatures. Specifically, is the non-ergodicity transition avoided in the theory from Eq. (5-8)? This paragraph should be rewritten for clarity and avoid confusion.

This also appears in the discussion section as an exact result for the validity of the theory (that is in comparison with simulation). What the authors actually did is test how various approximations compare with respect to the results before these approximations are imposed for analytical traceability. Thus, the work holds even if Fig. 1 gives poor comparison at lower temperatures.

(6) Last paragraph of Sec. 3.4: I am confused regarding this discussion as well. The factorization approximation uses the Gaussian nature of density fluctuation. Moreover, chi_4(t) diverges at the MCT transition that controls the entire physics within MCT treating glass transition as an ordinary phase transition. Could this be the source of the failure of MCT?

This is also related to the discussion on Page 12, the second paragraph, and the last paragraph in Sec. 4. Can we neglect the off-diagonal terms when the density fluctuations are Gaussian? Higher order closure relations seem to work better for MCT, this also seems to suggest that the assumption of the Gaussian nature of the density fluctuations might be a problem.

I think this is an important and challenging problem that the authors have studied in this work. I would have liked to see a much lower range of temperature than the one studied in this work. T=1.0 is nearly the high-T liquid. A set of comparisons at much lower temperatures of each of these approximations can be insightful. The authors do discuss the way forward for a more quantitative dynamical theory. I think a discussion on how to avoid the non-ergodicity transition might also be useful.
  • validity: ok
  • significance: ok
  • originality: good
  • clarity: good
  • formatting: reasonable
  • grammar: good

Author:  Ilian Pihlajamaa  on 2023-10-11  [id 4034]

(in reply to Report 1 by Saroj Nandi on 2023-10-02)
Category:
answer to question
correction

We thank the referee for their kind words and their recommendation. We are happy they agree our work is important and suitable for a broad audience. Below we address their comments.
(1) We have changed it to “Predicting the microscopic dynamics of dense and supercooled liquids stands out as a large unsolved problem in classical physics”, where it used to say “… stands out as one of the largest … ”.
(2) We have added the references. Additionally, as Referee 2 suggested, we have added Szamel et al., PRE (2015) and Szamel PRE (2016) when referring to MCT for active systems.
(3) We have toned down the statement. It now reads “The above observations lead to the belief that the breakdown of MCT at $T_{MCT}$ coincides with a physical change in the behavior of glassy liquids”.
(4) We agree this is an important point, which is the reason that we have mentioned it multiple times in the introduction in the original draft
a. “In a field-theoretic setting, MCT can be derived as a self-consistent one-loop resummation [52, 54–56]”
b. “Various kinetic-like approaches have also led to the same MCT equations [54, 59]”
To make it more clear in the outlook section of the conclusion that we are solely speaking about projector-operator based approaches, we have adapted the sentence to “One can envision several routes toward a more quantitative dynamical theory of the glass transition [based on the Mori-Zwanzig approach],” where we added the part between square brackets.
(5) Equations (5) and (8) constitute a coupled set of exact equations. No approximation has been made in their derivation. We are not the first to present them, see for example Eq. (21) in [J. Chem. Phys. 119, 12063–12076 (2003)], and related works (such as [J. Chem. Phys. 144, 184105 (2016)], [J. Chem. Phys. 144, 184104 (2016)], [J. Chem. Phys. 151, 084503 (2019)]). Indeed if one is in possession of exact data for K_omega and W, and both of which go to zero in the long-time limit, it is straightforward to show in the Laplace domain that the intermediate scattering function also decays to zero.
Unfortunately, the inversion of eq 8 becomes numerically ill-conditioned as t increases to very large values, amplifying any numerical noise of K_omega and W, which are determined from simulations. This is the cause of the tiny deviation for the lowest temperature studied.
(6) This point is split up into three and as such we address them separately:
a. The point we are making is that while the density fluctuations are non-Gaussian (otherwise chi_4 would be zero), the error that one makes when factorizing is vanishingly small. Specifically, factorizing <rho1 rho2 rho1(t) rho2(t)>/N^2 - <rho1rho1(t)>/N * <rho2 rho2(t)>/N, is equal to chi_4(k1, k2)/N by definition. The susceptibility here is an intensive, finite quantity at all temperatures measured, and while it does diverge as predicted by MCT at the transition, it does so only precisely at the transition, meaning that our statement remains true at any temperature T \neq T_MCT. This means that the factorization is exact for all intents and purposes.
The fact that MCT predicts a divergent chi4 cannot be the source of MCT’s failure, since, in the theory, chi4 does not couple back to F. In an exact version of MCT, where the transition is avoided, so too will the divergence of chi4, by consequence.
b. Indeed, if density fluctuations were Gaussian, then by Wick’s probability theorem, one can factorize any even product of pairs. Since, in the off-diagonal case, none of the resulting terms satisfy momentum conservation they may all be neglected.
c. We also would like to have studied lower temperatures (this is in fact ongoing work). Lower temperatures however pose two main difficulties that render this task both technically far more challenging, and would make the current manuscript more difficult to read and hence obscure the main message.
i. In order to study lower temperatures, we need to apply the same methods to mixtures such that crystallization in the supercooled regime is avoided. That means that all expressions would gain multiple additional indices to label the different particle species, and instead of a single curve per approximation, we would need to compute and compare at least three.
ii. The current method that we use for inverting eq (8) is not well-conditioned, and will fail at lower temperatures (even though the identity itself is exact). So, new, more complex methods need to be developed to avoid this.
These are the reasons that we prefer to present our results first in the simpler (but arguably less interesting) liquid regime, and only later do the same for the more complicated case of mixtures at supercooled conditions.
We hope we have answered the referee’s questions satisfactorily.

Attachment:

diff.pdf

---

## Round 1 · Referee Report · Anonymous (Referee 2) · 2023-10-3

Strengths

  1. Very clear examination of approximations used in mode-coupling theory of glassy dynamics.

Weaknesses

  1. The approximations were examined only for very slightly supercooled fluids (which do not exhibit the two-step decay of the intermediate scattering function). Thus, it is not immediately clear whether the conclusions are applicable for fluids near the mode-coupling crossover.

Report

The paper is interesting, useful and clearly written (admittedly, this referee is vary familiar with the subject and thus may overlook some not-clear-to-newcomers parts). I recommend that it is accepted once the authors consider the following comments and modify the paper accordingly.

The only major comment is concerned with a quite delicate and often ignored issue of "density doublets". It seems to me that the authors do not appreciate the fact that it is not obvious that "the inverse four-point structure factor" exists. They may find it useful to consult the discussion following Eq. (12) and comment 27 in Andersen, JPC B 106, 8326 (2002), and the discussion in Mazenko, PRA 9, 360 (1974), cited by Andersen. Basically, one should first define the part of pair density orthogonal to the one-particle density. Then, the correlation function of this pair density has a well-defined inverse. And this inverse then can be used to derive Eqs. (13-14).

It follows that the projection on "density doublets" is really the projection on the part of the pair density orthogonal to the one-particle density and, in fact, it is an exact transformation (for a Brownian system) rather than an approximation. The approximation is to use the diagonal (or factorized) part of the inverse.

Minor comments:
a) The first mode-coupling-like theory for systems of self-propelled particles was proposed by Szamel et al., PRE 91, 062304 (2015).
b) It is confusing when the authors refer to K_{\Omega^\dagger} as "the kernel with projected dynamics" or "the projected kernel". This kernel evolves with the original, un-projected dynamics.
c) The authors' emphasis on the separation/difference between factorization and diagonal approximation is, in my humble opinion, overdone. For example, if one considers only those diagrammatic contributions to the memory function that separate when the vertices are removed, one gets and approximate expression that is both diagonal and factorized.
d) It is indeed true that generalized mode-coupling approaches impose additional structure on the diagonal part of the pair-pair correlation function. Thus, they remove some diagrammatic contributions included in the standard mode-coupling theory. It is interesting but no unheard of that including fewer diagrams leads to a more accurate approximate theory. For example, PY approximation includes fewer diagrams than HNC but for hard spheres PY is more accurate than HNC.
e) While it may seem ironic that "the most attempts have been made in the past to circumvent" factorization and convolution approximations, the authors certainly appreciate the fact that including off-diagonal terms is numerically quite challenging.
f) Off-diagonal contributions for the simpler problem of a big Brownian particle in a small particle bath was considered by Masters and Madden, JCP 74, 2450 (1981) and then generalized by Charbonneau et al., JCP 148, 224503 (2018).
  • validity: high
  • significance: high
  • originality: high
  • clarity: high
  • formatting: good
  • grammar: excellent

Author:  Ilian Pihlajamaa  on 2023-10-11  [id 4035]

(in reply to Report 2 on 2023-10-03)
Category:
answer to question
correction

We are delighted by the kind words of the referee. Below we try to address their questions and comments.
Firstly, the referee rightly points out that we have been imprecise in defining our projection operators. We have tried to rectify it in the revised version of the manuscript. Please consult the first paragraph of section 3.2, and the last paragraph of appendix A for the changes we have made to the text. We are attaching a pdf where the differences are marked. We hope that the referee agrees with the changes.
Regarding the minor comments:
a) We apologize for accidentally having missed this important reference. As mentioned in the response to referee 1, we have further revised the referencing for self-propelled particles.
b) We agree, and have changed these to be unambiguous.
c) We understand the point the referee is making. This is also why we included the sentence “We note that in some works, the dynamic diagonalization approximation and factorization are collectively referred to as the factorization approximation, because the factorization of a four-point function implies its diagonalization.” We understand that from a diagrammatic, and perhaps more broadly from a field-theoretic point of view, these should not be separated. However, this has not been as clear to us for a long time. Confusion about this quickly leads to the situation where one has a fully offdiagonal four-point theory, that one diagonalizes for computational reasons without realizing that this immediately nullifies the entire point of the exercise, because it reduces to a two-point theory by factorization. In this light, we prefer to be overly cautious, while perhaps semantically/fundamentally inconsistent/incorrect.
d) Yes, interestingly, G. Szamel makes the same point in [AIP Conf. Proc. 982, 62–69 (2008)], where he writes “It can be noted that, at least as far as the location of the ergodicity breaking transition is concerned, the generalized mode-coupling theory is superior to the standard theory. This occurs in spite of the fact that the former theory re-sums fewer diagrams. This fact is not unprecedented: for short-range repulsive interaction potentials the Percus-Yevick integral equation for the equilibrium pair-correlation function is more accurate than the HNC equation in spite of including fewer diagrams.”
While we agree with the main point, we would like to add that in the case of the Ornstein-Zernike closure theory, one can explain (ad hoc) why it should be so (see the discussion below eq 4.6.17 in the book of Hansen & McDonald for example). For the case of (G)MCT, it is not clear to us why the omission of a particular set of diagrams introduces a more complete cancellation of errors than when they are retained.
e) Of course! We did not mean to suggest that we do not understand why the most effort has gone into these particular approximations.
f) We have added the references.

Attachment:

diff_2R8mFuj.pdf

---

## Editorial Decision

resubmitted